# p-S6 as a Prognostic Biomarker in Canine Oral Squamous Cell Carcinoma

**DOI:** 10.3390/biom12070935

**Published:** 2022-07-04

**Authors:** Leonor Delgado, Paula Brilhante-Simões, Fernanda Garcez, Luís Monteiro, Isabel Pires, Justina Prada

**Affiliations:** 1Pathology Department, INNO Serviços Especializados em Veterinária, 4710-503 Braga, Portugal; paula.simoes@iucs.cespu.pt; 2UNIPRO, Oral Pathology and Rehabilitation Research Unit, University Institute of Health Sciences (IUCS), CESPU, 4585-116 Gandra, Portugal; fernanda.garcez@cespu.pt (F.G.); luis.monteiro@iucs.cespu.pt (L.M.); 3Veterinary Medicine Department, University Institute of Health Sciences (IUCS), CESPU, 4585-116 Gandra, Portugal; 4Medicine and Oral Surgery Department, University Institute of Health Sciences (IUCS), CESPU, 4585-116 Gandra, Portugal; 5Department of Veterinary Science, University of Trás-os-Montes and Alto Douro, 5001-801 Vila Real, Portugal; ipires@utad.pt (I.P.); jprada@utad.pt (J.P.); 6CECAV-Veterinary and Animal Research Center, University of Trás-os-Montes and Alto Douro, 5001-801 Vila Real, Portugal

**Keywords:** oral cancer, p-mTOR and p-S6 expression, prognostic markers, survival

## Abstract

Scarce information exists on the role of mTOR pathway proteins and their association to aggressiveness and prognosis of patients with canine oral cancers. We aimed to investigate the activated form of mTOR and its downstream S6 protein in canine oral squamous cell carcinoma (OSCC), and to evaluate potential associations between protein expression and clinic-pathologic variables and survival. For that we analysed p-mTOR and p-S6 protein expression by immunohistochemistry in 61 canine OSCCs. Multivariate analysis was conducted to examine their role in patients’ cancer-specific survival (CSS). p-mTOR and p-S6 expression were present in almost all cases. High-expression of p-mTOR was observed in 44 (72.1%) cases using extent score and 52 (85.2%) cases using intensity score. For p-S6, high expression was observed in 53 (86.9%) cases using extent score and in 54 (88.5%) cases using intensity score. An independent prognostic value for p-S6 extension (*p* = 0.027), tumour stage (*p* = 0.013) and treatment (*p* = 0.0009) was found in patients’ CSS analysis. Our data suggest that p-mTOR and p-S6 proteins are commonly expressed in canine OSCC and p-S6 expression is correlated with poor CSS in dogs with OSCC. More studies should be performed to identify possible therapeutic targets related with mTOR pathway for these patients.

## 1. Introduction

The oral cavity is a common site for malignant tumours, accounting for 5% to 7% of all canine cancers. Oral squamous cell carcinoma (OSCC) is the second most common malignant oral neoplasm (17% to 25%) in dogs, after melanoma (30% to 40%) and followed by fibrosarcoma (8% to 25%) [1,2].

Many similarities exist between the World Health Organisation classification scheme of oral cancers in human and canine tumours. Recently, canine OSCC were divided in canine oral papillary squamous cell carcinoma and conventional squamous cell carcinoma [3], with papillary subtype usually without metastasis, amenable to curative treatment and having a better outcome [4,5]. Other less common subtypes are also described including basaloid SCC, spindle cell carcinoma and adenosquamous carcinoma [6].

Risk factors for human oral cancer are well stablished and include tobacco consumption, alcohol misuse and other factors such as HPV infection in some oral sites (e.g., oropharynx) [7,8,9,10,11]. In dogs, aetiology and risk factors are poorly documented and the existing ones have weak scientific evidence. Dogs and other companion animals share the human environment and are exposed to many of the same carcinogens as man (e.g., as second-hand smoking). Second-hand smoking has been demonstrated as a risk factor for oral cancer in humans with strong evidence [12,13], suggesting a possible role also for dogs. Nevertheless, many canine cancers occur naturally and could be related with intrinsic genetic alterations. All of them, could be used as a potential model in the study of carcinogenesis of these neoplasms not only in dogs but also in humans [14].

The PI3K/AKT/mTOR intracellular signalling pathway plays a crucial role in several biological and metabolic processes, such as cellular growth and survival, proliferation, protein synthesis, angiogenesis and glucose metabolism [15,16,17,18,19,20].

mTOR can combine with different proteins and form two distinct complexes: mTORC1 and mTORC2. These complexes have different downstream effectors and physiological functions: mTORC1 effectors are 4EBP1 and S6K1, a 70-kDa ribosomal S6 kinase which participate in the subsequent phosphorylation of S6 ribosomal protein (also called p-S6) with important functions on cellular growth, proliferation and survival. mTORC2 can phosphorylate protein kinase C-alfa (PKC-α) and AKT (Ser 473) and regulates the actin cytoskeleton and thus cell migration [15,21]. This pathway is often dysregulated in many human cancers, such as thyroid, breast, ovarian, lung, gastric and oral cancers [15,16,17,18,22,23]. Previous studies [17] demonstrated that phospho-mTOR (p-mTOR) is associated with an adverse outcome in OSCC indicating that this marker may serve to identify patients at high-risk of worse prognosis. Additionally, p-S6, a downstream target of p-mTOR, has been shown to be a biomarker of mTOR activity, specially using immunohistochemistry method, and could be studied as a prognostic biomarker related with PI3K/mTOR signalling [24]. Research on this pathway in canine tumours is scarce. Some studies in canine mammary carcinomas [25], prostatic carcinomas [26], haemangiosarcomas [27] and in canine cell lines of B-cell lymphoma, glioma and mast cell tumour cells [28] have reported that mTOR pathway and its downstream effectors are activated in dogs, suggesting the need for research directed to the potential use of these proteins not only as prognostic biomarkers but also as a target for molecular therapies aimed at these biomarkers in canine SCC, as has been studied in human cancers [24,29,30,31].

The aim of the present study was to evaluate p-mTOR and p-S6 expression in OSCC and relate them to clinic, pathologic and prognostic features in a cohort of dog patients.

## 2. Materials and Methods

### 2.1. Patients and Tissue Specimens

Formalin-fixed and paraffin-embedded tissue samples of OSCC from 61 dog patients, obtained consecutively from Pathology Laboratory—INNO, between January 1st of 2010 and December 31st of 2017, were included in this retrospective study. The study was approved by the INNO lab administration board (nº INNO/2021/01) and was performed according to the Helsinki declaration. As inclusion criteria we included cases from several anatomical sites of the oral cavity (ICD 10: C00-06) with a confirmed histopathology diagnosis of oral squamous cell carcinomas. Cases with a history of previous treatment for oral cancer or cases without any histopathological confirmation were excluded. The variables analysed (including information’s obtained from Portuguese veterinary hospitals and clinics of respective samples; by direct contact, email or repeated telephone interviews with referring veterinarians) comprised age, gender, breed, location of the lesion, size of lesion, histopathological diagnosis, histopathological grade, presence of bone and vascular invasion, number of mitosis, presence of necrosis, presence of nuclear pleomorphism and lymphocytic infiltration and follow-up.

Age was grouped in two groups, one comprising “young and adult” dogs with age below 7 year-old and other above 7 year-old classified as “senior to geriatric” dogs [32]. Breeds were grouped into small breeds, medium breeds, large breeds and mixed breeds as applied in a previous study [2]. Tumour location was grouped into mouth not otherwise specified (MNOS), gingiva, tongue, tonsil/oropharynx, palate and other locations [2].

Histopathological diagnosis was performed based on World Health Organisation (WHO) histological classification of tumours of the alimentary system of domestic animals (2003) [33]. Histopathological grade was performed according to the Anneroth’s et al. (1987) [34] multifactorial grading system. According to this system, six morphological parameters were evaluated: degree of keratinisation; nuclear pleomorphism; number of mitosis/high power field (hpf); pattern of invasion; stage of invasion; and lymphoplasmacytic infiltration. Degree of keratinisation was evaluated by observing the percentage of tumour keratinised cells as follows: I—>50% cells keratinised; II—20–50% cells keratinised; III—5–20% cells keratinised; and IV—0–5% cells keratinised. Nuclear pleomorphism was evaluated as: I—little nuclear pleomorphism; II—moderately abundant nuclear pleomorphism; III—abundant nuclear pleomorphism; IV—extreme nuclear pleomorphism. Number of mitoses were evaluated in 1 hpf as: I—0 to 1 mitosis/hpf; II—2 to 3 mitosis/hpf; III—4 to 5 mitosis/hpf; IV—>5 mitosis/hpf. Pattern of invasion was evaluated as: I—pushing, well delineated infiltrating borders; II—infiltrating, solid cords, bands and/or strands; III—small groups or cords of infiltrating cells; IV—marked and widespread cellular dissemination in small groups and/or in single cells. Stage of invasion was classified as: I—corresponding to carcinoma-in situ and/or questionable invasion; II—distinct invasion, but involving lamina propria only; III—invasion below *lamina propria* adjacent to muscles, salivary gland tissues and periosteum; IV—extensive and deep invasion replacing most of the stromal tissue and infiltrating the jawbone. Lymphoplasmacytic infiltration was evaluated as: I—marked; II—moderate; III—slight; IV—none. For Anneroth’s overall grade score [34], the sum of the scores (with each parameter graded in the 4 categories) originate the following groups: grade I—6 to 12 points; grade II—13 to 18 points; and grade III—19 to 24 points. Cases were also graded according to Bryne’s et al. (1989) [35] score. In this system, the number of mitosis and stage of invasion is omitted from the Anneroth’s grading system, while the rest of the four parameters mentioned above were measured in the deepest invasive margins, and graded similarly. The sum of scores were grouped as follows: grade I—4 to 8 points; grade II—9 to 12 points; and grade III—13 to 16 points.

Presence of vascular and bone invasion was categorised as present or absent by histopathological evaluation [17].

We classified tumour stage by combining information of tumour size and presence of invasion from clinical, imagiological, macroscopic or microscopic analysis adapted from the human classification of AJCC Cancer Staging Manual, 8th edition [36] and corroborated by the authors of tumours of the alimentary tract [37].

### 2.2. Immunohistochemistry Methodology and Evaluation

The expression of p-mTOR and p-S6 proteins was evaluated by immunohistochemistry on 3-µm whole-tissue sections, using the antibodies: anti-phospho mTOR at Ser2448 (rabbit monoclonal antibody, clone 49F9; Cell Signaling Technology, Beverly, MA, USA), and anti-phospho S6 Ribosomal Protein at Ser235/236 (rabbit polyclonal antibody; Cell Signalling Technology, Beverly, MA, USA). These primary antibodies were previously validated in canine tissues [25,28,38]. Briefly, tissue sections were dewaxed in xylene and hydrated through a decreasing series of alcohol concentrations, followed by antigen-retrieval treatment (citrate buffer 0.01 M pH 6.0 for both antibodies) at high temperature (water bath, 30 min at 98 °C). After blocking for nonspecific binding, primary antibody was added to the sections in an optimised dilution (p-mTOR 1/150; p-S6 1/300) and incubated for 60 min at room temperature. The primary antibodies were detected using a standard peroxidase-labelled dextran polymer for visualisation with diaminobenzidine as chromogen (NovoLink Polymer Detection System; Novocastra, Leica Biosystems Newcastle, UK), according to the manufacturer’s instructions. Then, sections were lightly counterstained with Gill’s haematoxylin and cover-slipped. Negative (omission of primary antibody) and positive controls (a breast carcinoma) were used in each staining run.

All samples were evaluated independently by two observers (LD and LM) blinded to clinic and pathologic characteristics and using a ZEISS AxioLab A1^®^ microscope (Carl Zeiss Microscopy GmbH, Jena, Germany), equipped with a ZEISS Axiocam 105 colour ^®^ and ZEISS Zen2^®^ software (Carl Zeiss Microscopy GmbH, Jena, Germany). The discordant cases were reviewed by a third observer (JP) and discussed together to achieve a final score.

Phospho-mTOR and phospho-S6 immunoreactivity was assessed semi-quantitatively on the basis of the extent of labelling (percentage of stained tumour cells, in 10 high power field, considering a minimum count of 150 tumour cells per field) (considering cytoplasm staining) and was scored as follows: 0, 0–9% of tumour cells labelled; 1+, 10–24% of tumour cells labelled; 2+, 25–49% of tumour cells labelled; 3+, 50–74% of tumour cells labelled; or 4+, 75–100% of tumour cells labelled and intensity of staining of p-mTOR and p-S6 as 0 (negative), 1 (weak), 2 (moderate) and 3 (strong). For data analysis of each biomarker, we calculate cut-off values using ROC curves (dead as event) that were used to distinguish low and high-expression cases: p-mTOR extension, 0/1+/2+ vs 3+/4+; p-S6 extension, 0/1+ vs 2+/3+/4+; and intensity scores for p-mTOR and p-S6, absent/weak vs moderate/strong intensity.

### 2.3. Statistical Analysis

Statistical analysis was performed using IBM SPSS Statistics version 27.0 software (IBM Corporation, Armonk, NY, USA). The associations of p-mTOR and p-S6 expression with clinic and pathologic parameters variables were assessed by chi-square test.

For univariate survival analysis, Kaplan–Meier and log-rank tests were performed. For multivariable analysis, Cox proportional hazards model was used, considering variables with significant result in the univariate analyses. Cancer-specific survival (CSS) for each dog was defined as the time interval (months) between histologic diagnosis of OSCC and death as a result of oral cancer. Cases of loss of follow-up but with previous information of evidence of persistent/or advanced or disseminated disease or if they had died or been euthanised for reasons related to their tumour were considered as dead by tumour (using date of dead or their last contact) [39,40]. Survival times of the patients who were still alive at the end of follow-up or who died from other causes not associated with oral cancer were censured at the last date that they were seen alive or at the date of death, respectively.

For all tests, the level of significance was set at probabilities of *p* < 0.05.

## 3. Results

The study group included 61 canine patients, 33 male (54.1%) and 28 female dogs (45.9%), with ages ranging from 1 to 15 years, with an average age of 10.9 ± 2.7 years. Further clinical and pathologic characteristics are presented in the Table 1.

### 3.1. p-mTOR Expression Analysis

The p-mTOR expression was classified as 0–9% in 4 (6.6%), 10–24% in 3 cases (4.9%), 25–49% in 10 (16.4%), 50–74% in 18 (29.5%) and 75–100% in 26 cases (42.5%). For data analysis, p-mTOR was divided regarding extent score into low expression (from 0 to 49%) in 17 (27.9%) cases and high expression (from 50 to 100%) in 44 (72.1%) cases (Table 2).

Regarding intensity, 9 cases (14.8%) were classified as weak, 25 (41%) as moderate and 27 (44.3%) as strong intensity cases and grouped as negative/weak intensity (9; 14.8%) and moderate/strong intensity (52; 85.2%) cases for statistical analysis (Table 2). Most of the staining was detected in cytoplasm and cell membrane of tumour cells (53; 86.9%) and in nucleus in 3 cases (4.9%) and only in cell cytoplasm of 5 (8.2%) cases. Most of the staining was homogenously distributed over the tumour islands in 49 cases (80.3%), in central part of tumour islands in 8 cases (13.1%) and in the periphery of tumour islands in 4 cases (6.6%) (Figure 1).

When we analysed the association between p-mTOR and clinic/pathologic characteristics, a significant association was observed between p-mTOR extent score with pattern of tumour invasion when categorised by I vs II + III + IV grades (*p* = 0.03) and between p-mTOR intensity score expression and number of mitosis (*p* = 0.005) (Table 2). Tumours with a high pattern of invasion were less likely to have a high p-mTOR extent score. Moreover, tumours with a high number of mitoses were less likely to have a high p-mTOR intensity score.

### 3.2. p-S6 Expression Analysis

The p-S6 expression was detected in almost all of the cases (60; 98.4%) and classified as 0–9% in 2 cases (3.3%), 10–24% in 6 cases (9.8%), 25–49% in 7 (11.5%), 50–74% in 20 (32.8%) and 75–100% in 26 cases (42.6%). For data analysis, p-S6 expression was divided 0–24% in 8 (13.1%) cases and high-expression in 53 (86.9%) cases (Table 3).

Regarding intensity, one case was negative (1.6%), 6 cases (9.8%) were classified as weak, 25 (41%) as moderate and 29 (47.5%) as strong intensity cases and grouped as negative/weak intensity (7; 11.5%) and moderate/strong intensity (54; 88.5%) cases for statistical analysis (Table 3). Staining was observed in cell cytoplasm of all detected cases and in some cases (28; 45.9%) also in the membrane. The expression distribution within the tumour islands was homogeneous in 44 (73.3%) cases, or was predominantly detected in their central part in 7 cases (11.7%) or at the periphery in 9 cases (15%) (Figure 2).

A significant association was observed between p-S6 intensity expression and presence of necrosis (*p* = 0.042) where tumours with necrosis were less likely to have a high p-S6 intensity score (Table 3).

### 3.3. Correlation between Biomarkers

Tumours with high expression of p-mTOR tend to possess a higher expression of p-S6, although this was observed with statistically significant correlation only in extent score analysis (*p* < 0.001) even when categorised in two categories (*p* = 0.001). Correlation of both biomarkers using intensity score was not significantly evident (*p* = 0.237) even when categorised in two categories (*p* = 0.281).

### 3.4. Analysis of Cancer-Specific Survival

At the end of the study, from 50 patients with information concerning survival, 5 patients (10%) were alive without oral cancer, 6 patients (12%) were alive with oral cancer, 33 patients (66%) had died as a result of oral cancer and 6 (12%) died from other causes. Cancer-specific survival rate corresponded to 30.5% at 1-year of follow-up and 22% at 2-years of follow-up. The follow-up mean for all patients was 6.8 ± 1.5 months.

In univariate analysis, tumour stage (*p* = 0.001), histological type (*p* = 0.013), pattern of invasion (*p* = 0.011), stage of invasion (*p* = 0.009), treatment (*p* = 0.048) and p-S6 extension (*p* = 0.023) were statistically associated with cancer-specific survival (Table 4 and Table 5; Figure 3 and Figure 4).

In multivariate analysis for CSS, we found an independent prognostic value for treatment, p-S6 extension and tumour stage, where dogs with tumours treated with palliative or support approach (HR 1.949; 95% CI, 1.178–3.223, *p* = 0.009), with tumours with high p-S6 expression (HR 11.577; 95% CI, 1.316–101.826, *p* = 0.027) or tumours with advanced tumour stage (HR 5.005; 95% CI, 1.405–17.829, *p* = 0.013) had more risk of dying comparing with the respective other categories of each variable (Table 6).

## 4. Discussion

This study provides the first analysis of protein expression associated with the PI3K/AKT/mTOR pathway in canine oral squamous cell carcinoma, correlating them with clinic and pathologic parameters. Based on the present sample, we found that almost all cases expressed the activated (i.e., phosphorylated) form of mTOR, with the majority of the cases revealing a high expression of this marker (72.1%). In one study including canine mammary carcinomas, p-mTOR labelling was found in 78% of the cases (n = 45) [25] and in another study on canine prostatic carcinomas p-mTOR occurred at higher levels (more than 75%) [26]. By contrast, in canine hemangiosarcomas only 35% of samples had weak to moderate expression of p-mTOR [27]. These variations may be associated to the different locations of the tumours, with our results suggesting high expression in canine OSCC.

Increasing attention has been paid to the mTOR pathway specially because it could be a potential target for cancer therapy in both dogs and humans [15,16,21,23,26,27,41,42,43,44,45,46]. This is directed to decrease the activity of tumour cell using single molecules such as everolimus, curcumin or in combination with other therapies [47,48,49], to decrease the resistance to chemotherapy by mTOR inhibition (e.g., temsirolimus) [50] and radiotherapy [51] and interestingly also to prevent oral cancer, by its use in human oral potentially malignant disorders (OPMD), where the use of molecules such as metformin, or other molecules decreased these premalignant lesions with decreased mTOR activity [24]. Our results on mTOR expression (mostly high expression) suggest that canine OSCC should be a candidate for further investigation of target therapy directed toward mTOR.

No information on the association of p-mTOR expression with clinical/pathologic variables in canine oral cancer exists so far, to our knowledge, which makes comparisons of the present results difficult. In the present study, we found significant association only between p-mTOR extent score and pattern of tumour invasion (when grouped I + II vs III + IV) and between p-mTOR intensity score expression and number of mitosis, although in a different association direction than the expected. This can be related with the present sample constitution, by the participation of other phosphorylation sites than that evaluated in this study or an indirect effect of other regulation mechanism by other genes/proteins or pathways. In other canine cancers, associations with clinic and pathologic variables were not common. In one study by Delgado et al. (2015) [25] in canine mammary carcinomas no significant relationship was found between p-mTOR cytoplasmic expression and histological type or grading of carcinomas, degree of tubular formation, anisokaryosis, mitotic activity or lymph node metastasis. In another study on canine prostatic carcinoma, p-mTOR protein level was positively correlated with higher Gleason score [26]. In human OSCC, p-mTOR was not significantly related with any of the clinical and pathologic variables [17].

Proteins in the mTOR pathway could indicate some information on prognosis of patients with head and neck cancers [23]. In one study on human OSCC, tumours with high expression of p-mTOR had lower overall survival with an independent effect noted in multivariate analysis [17]. In the present study p-mTOR expression was not statistically related with CSS. By contrast, tumours with advanced tumour stage, high pattern or high stage of invasion, conventional histological type tumours, or higher rate of squamous differentiation were related with worse survival rates. mTOR deregulated status in canine carcinogenesis could be related with other several molecules with multidirectional functions, compromising the value of this protein as a biomarker of prognosis in these tumours. However, the p-S6 protein has been used as a biomarker of the function of mTOR in human cancers [24]. We initially hypothesised if this protein could be more specific as prognostic biomarker than mTOR. Mammalian TOR is a serine/threonine kinase which phosphorylates translation regulators such as the p70-S6 kinase that in turn phosphorylates the ribosomal protein S6, the most downstream target of this pathway [52]. In this view, an overexpression of p-S6 could indicate a deregulated activation of mTOR pathway. In the present sample, we found that the majority of cases reveal a high expression of p-S6 which is in line with that observed in canine cutaneous squamous cell carcinoma (cSCC) in Ressel et al., 2019 [52] study. Most of their cSCC samples (130/140, 92.85%) showed a high expression of p-S6, with many of the cells with a cytoplasmic staining located in the basal and parabasal cells layers, as observed in our samples of OSCC. Furthermore, and confirming our initial hypothesis, in the present study, p-S6 expression was related to CSS, where tumours with high expression had lower survival with an independent effect noted in multivariate analysis. To the best of our knowledge, the data presented here is the first report of both p-mTOR and p-S6 expression in canine OSCC, highlighting p-S6 as an independent prognostic factor in canine OSCC. Unfortunately, there is no other study evaluating p-S6 as prognostic biomarkers in canine OSCC. In human OSCC, Vicent et al. (2017) analysed the expression of p-S6 in a series of 125 human patients with OSCC and verified expression of p-S6 protein on either serine 235/236 or serine 240/244 in 83% and 88% tumours, respectively, and both of them were inversely and significantly associated with the tumour size and local infiltration, however no associations were found with survival outcomes [53].

Additionally, to these results, many interest is also directed to the possibility of using this molecule, as a target for anticancer therapy or as a biomarker of therapy response. On a study by Enjoji et al. (2015) [54] on canine melanoma cell lines phosphorylation level of p70, S6 kinase was decreased by SET (an endogenous inhibitor for PP2A, a serine/threonine phosphatase) knockdown. The results of this study demonstrated the potential therapeutic application of SET inhibitors for canine melanoma. On a similar manner, S6 kinase inhibitors may be used on canine squamous cell carcinomas. On two others similar studies, one by Kent et al. (2009) [43] in canine melanoma cell lines and the other in canine osteosarcoma cell lines treated with rapamycin, both resulted in a reduction of phosphorylated mTOR expression and phosphorylated p70S6K expression. These data support the molecular basis for using mTOR inhibitors as an antineoplastic approach in canine cancer. Another study (Hsu C, et al. (2018)) confirmed the effect of NVP-BEZ235 (dactolisib), a dual PI3K/mTOR inhibitor, on human OSCC cells, showing a new strategy for controlling the proliferation, migration and invasion of OSCC cells using a phopho-p70S6K inhibitor [55].

We acknowledge some limitations in our study, including its retrospective nature, small sample of patients, short follow-up time and incomplete clinical information that led to the exclusion of some cases that could have participated in the study. However, including multivariate analysis we controlled some potential confounding variables showing a promising role of these biomarkers, specially p-S6, in canine OSCC.

## 5. Conclusions

Our data suggest that p-mTOR and p-S6 proteins are commonly expressed in canine OSCC, and p-S6 expression is correlated with poor CSS in dogs with OSCC. Further understanding with regard to the role of the mTOR pathway in canine OSCC may provide an improved insight of oral tumorigenesis and may open up new treatment possibilities for these tumours.

## Figures and Tables

**Figure 1 biomolecules-12-00935-f001:**
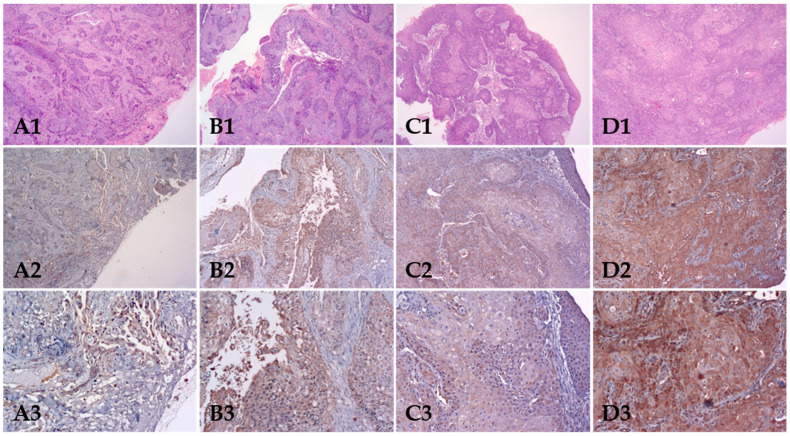
Immunohistochemical (IHC) staining of the p-mTOR in four cases of OSCC (**A**–**D**), showing low expression (extent and intensity) (case (**A2**,**A3**)), high extent and low intensity (**B2**,**B3**,**C2**,**C3**) and high expression (extent and intensity) (**D2**,**D3**), (**B2**,**B3**). Numbers 1, 2 and 3 correspond to magnification at 5× (HE), 10× (IHC) and 20× (IHC), respectively.

**Figure 2 biomolecules-12-00935-f002:**
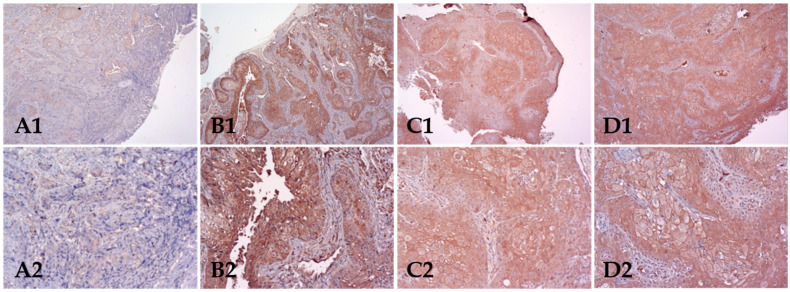
Immunohistochemical staining of the p-S6 in four cases of OSCC already observed in Figure 1, showing now low expression (extent and intensity) (**A**) and high expression (extent and intensity) of p-S6 (**B**–**D**). Numbers 1 and 2 correspond to magnification of 10× and 20×, respectively.

**Figure 3 biomolecules-12-00935-f003:**
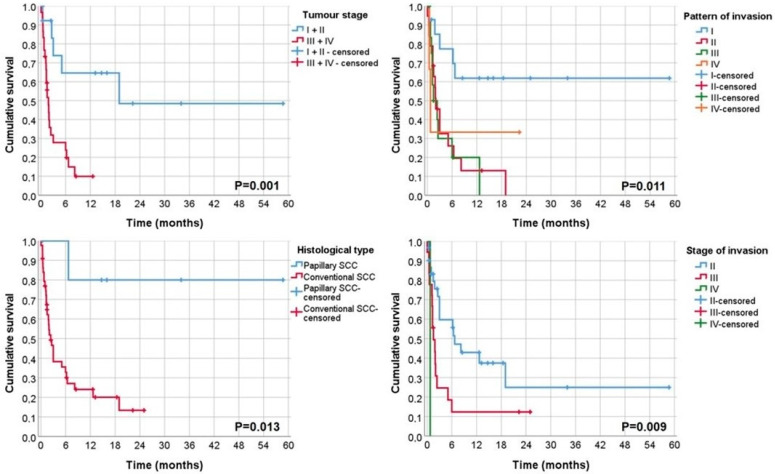
Kaplan Meier curve of CCS for tumour stage, pattern of invasion, histological type and stage of invasion.

**Figure 4 biomolecules-12-00935-f004:**
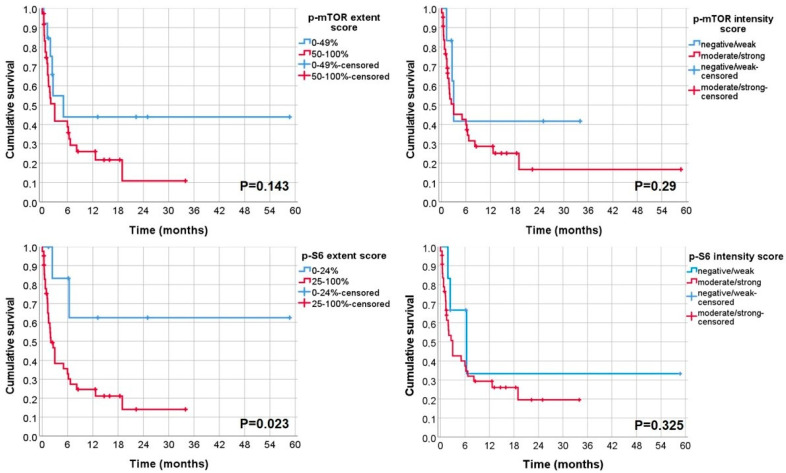
Kaplan Meier curves of CCS for p-mTOR and p-S6 extent and intensity scores.

**Table 1 biomolecules-12-00935-t001:** Patient characteristics (n = 61).

Variables	N (%)
**Gender**	
Female	28 (45.9%)
Male	33 (54.1%)
**Age**	
<7 year-old	6 (9.8%)
≥7 year-old	55 (90.2%)
**Breed (* n = 53)**	
Small	10 (18.9%)
Medium	5 (9.4%)
Large	12 (22.6%)
Mixed	26 (49.1%)
**Tumour Location**	
Mouth (NOS)	14 (23%)
Gingiva	20 (32.8%)
Tongue	12 (19.7%)
Oropharynx (including Tonsils)	8 (13.1%)
Palate	7 (11.5%)
**Histological type**	
Papillary SCC	10 (16.4%)
Conventional SCC	51 (83.6%)
**Anneroth´s histological grade**	
Well differentiated	20 (32.8%)
Moderate differentiated	41 (67.2%)
Poor differentiated	0 (0%)
**Bryne´s histological grade**	
Well differentiated	26 (42.6%)
Moderate differentiated	33 (54.1%)
Poor differentiated	2 (3.3%)
**Pattern of Invasion**	
I—Pushing, well delineated infiltrating borders	20 (32.8%)
II—Infiltrating, solid cords, bands and/or strands	22 (36.1%)
III—Small groups or cords of infiltrating cells	16 (26.2%)
IV—Marked and widespread cellular dissemination in small groups and/or in single cells	3 (4.9%)
**Stage of Invasion**	
I—Carcinoma-in-situ and/or questionable invasion	0 (0%)
II—Distinct invasion, but involving lamina propria only	39 (63.9%)
III—Invasion below lamina propria adjacent to muscles, salivary gland tissues and periosteum	21 (34.4%)
IV—Extensive and deep invasion replacing most of the stromal tissue and infiltrating the jawbone	1 (1.6%)
**Bone invasion**	
Absent	51 (83.6%)
Present	10 (16.4%)
**Vascular invasion**	
Absent	56 (91.8%)
Present	5 (8.2%)
**Tumour stage (* n = 51)**	
I +II	20 (39.2%)
III + IV	31 (60.8%)
**Treatment (* n = 50)**	
Surgery	11 (22%)
Chemotherapy	4 (8%)
Palliative treatment/support	35 (70%)

Legend: NOS, not otherwise specified; SSC, squamous cell carcinoma; * information not available for analysis for some cases.

**Table 2 biomolecules-12-00935-t002:** Clinicopathological characteristics of the OSCC patients and its association with p-mTOR expression.

		p-mTOR Extent Score			p-mTOR Intensity Score		
Variables		Low	High	*p*-Value	Low	High	*p*-Value
**All cases**		17	44	-	9	52	-
**Gender**	Female	7	21	0.645	3	25	0.412
	Male	10	23		6	27	
**Age (years)**	<7	2	4	0.753	2	4	0.177
	≥7	15	40		7	48	
**Breed**	Small	3	7	0.680	0	10	0.524
	Medium	2	3		1	4	
	Large	2	10		2	10	
	Undetermined	9	17		5	21	
**Tumour Location**	Mouth (NOS)	4	10	0.806	2	12	0.123
	Gingiva	4	16		1	19	
	Tongue	4	8		4	8	
	Oropharynx	2	6		0	8	
	Palate	3	4		2	5	
**Histological type**	Papillary SCC	1	9	0.168	1	7	0.847
	Conventional SCC	16	35		8	45	
**Bone Invasion**	Yes	2	8	0.544	0	10	0.150
	No	15	36		9	42	
**Vascular Invasion**	Yes	3	2	0.094	2	3	0.097
	No	14	42		7	49	
**Histological grade (Anneroth)**	Well differentiated	5	15	0.727	2	18	0.465
	Moderate differentiated	12	29		7	34	
	Poor differentiated	0	0		0	0	
**Histological grade** **(Bryne)**	Well differentiated	7	19	0.644	4	22	0.836
	Moderate differentiated	10	23		5	28	
	Poor differentiated	0	2		0	2	
**Mitosis**	0–1/hpf	4	13	0.350	0	17	**0.005**
	2–3/hpf	4	18		1	21	
	4–5/hpf	7	9		6	10	
	>5/hpf	2	4		2	4	
**Squamous differentiation**	>50% keratinisation	5	4	0.215	3	6	0.301
	20–50% keratinisation	3	10		1	12	
	5–20% keratinisation	6	14		3	14	
	0–5% keratinisation	6	16		2	20	
**Nuclear Pleomorphism**	Few	1	7	0.508	1	7	0.980
	Moderately	9	18		4	23	
	Abundant	7	19		4	22	
**Lymphoplasmacytic** **infiltration**	Weak	2	16	0.108	1	1	0.381
	Moderate	7	17		5	19	
	Marked	8	11		3	16	
**Necrosis**	Yes	9	12	0.059	4	17	0.493
	No	8	32		5	35	
**Pattern of invasion ***	I	2	18	0.160	2	18	0.732
	II	9	13		4	18	
	III	5	11		3	13	
	IV	1	2		0	3	
**Stage of invasion ***	I	0	0	0.151	0	0	0.742
	II	8	31		5	34	
	III	9	12		4	17	
	IV	0	1		0	1	
**Treatment ***	Surgery	5	6	0.244	3	8	0.186
	Chemotherapy	1	3		0	4	
	Palliative treatment/support	7	28		3	32	
**Tumour stage**	I + II	8	11	0.125	4	15	0.241
	III + IV	7	25		3	29	

* Pattern of invasion also evaluated for I + II vs III + IV (*p* = 0.664 and *p* = 0.878 for extent and intensity scores respectively) and I vs II + III + IV (*p* = 0.030 and *p* = 0.465); stage of invasion also evaluated for I + II vs III + IV (*p* = 0.088 and *p* = 0.571); and treatment using the categorisation of treatment in surgery/ chemotherapy vs palliative treatment/support (*p* = 0.140 and *p* = 0.254). Significant *p*-values are indicated as bold numbers.

**Table 3 biomolecules-12-00935-t003:** Clinicopathological characteristics of the OSCC patients and its association with p-S6 expression.

		p-S6 Extent Score			p-S6 Intensity Score		
Variables		Low	High	*p*-Value	Low	High	*p*-Value
**All**		8	53	-	7	54	-
**Gender**	Female	3	25	0.609	4	24	0.526
	Male	5	28		3	30	
**Age (years)**	<7	1	5	0.785	1	5	0.674
	≥7	7	48		6	49	
**Breed**	Small	2	8	0.282	0	10	0.259
	Medium	0	5		1	4	
	Large	0	12		0	12	
	Undetermined	5	21		4	22	
**Tumour Location**	Mouth (NOS)	1	13	0.395	1	13	0.953
	Gingiva	4	16		3	17	
	Tongue	0	12		1	11	
	Oropharynx	2	6		1	7	
	Palate	1	6		1	6	
**Histological type**	Papillary SCC	1	9	0.750	1	9	0.873
	Conventional SCC	7	44		6	45	
**Bone Invasion**	Yes	2	8	0.481	2	8	0.355
	No	6	45		5	46	
**Vascular Invasion**	Yes	0	5	0.365	1	4	0.533
	No	8	48		6	50	
**Histological grade** **(Anneroth)**	Well differentiated	1	19	0.190	2	18	0.801
	Moderate differentiated	7	34		5	36	
	Poor differentiated	0	0		0	0	
**Histological grade** **(Bryne)**	Well differentiated	2	24	0.424	4	22	0.659
	Moderate differentiated	6	27		3	30	
	Poor differentiated	0	2		0	2	
**Mitosis**	0–1/hpf	2	15	0.416	2	15	0.960
	2–3/hpf	3	19		2	20	
	4–5/hpf	1	15		2	14	
	>5/hpf	2	4		1	5	
**Squamous differentiation**	>50% keratinisation	0	9	0.263	1	8	0.097
	20–50% keratinisation	2	11		4	9	
	5–20% keratinisation	1	16		1	16	
	0–5% keratinisation	5	17		1	21	
**Nuclear Pleomorphism**	Few	1	7	0.505	1	7	0.715
	Moderately	5	22		4	23	
	Abundant	2	24		2	24	
**Lymphoplasmacytic** **infiltration**	Weak	1	17	0.322	2	17	0.978
	Moderate	5	19		3	21	
	Marked	2	17		2	16	
**Necrosis**	Yes	5	16	0.073	5	16	**0.042**
	No	3	37		2	38	
**Pattern of invasion ***	I	2	18	0.783	2	18	0.908
	II	3	19		3	19	
	III	3	13		2	14	
	IV	0	3		0	3	
**Stage of invasion ***	I	0	0	0.914	0	0	0.870
	II	5	34		5	34	
	III	3	18		2	19	
	IV	0	1		0	1	
**Treatment ***	Surgery	2	9	0.837	0	11	0.315
	Chemotherapy	1	3		1	3	
	Palliative treatment/support	5	30		5	30	
**Tumour stage**	I + II	2	17	0.832	3	16	0.492
	III + IV	4	28		3	29	

* Pattern of invasion also evaluated using the categorisation of I + II vs III + IV (*p* = 0.677 and *p* = 0.876 for extent and intensity scores respectively) and I vs II + III + IV (*p* = 0.615 and *p* = 0.801); stage of invasion also evaluated using the categorisation of I + II vs III + IV (*p* = 0.928 and *p* = 0.661); and treatment using the categorisation of treatment in Surgery/Chemotherapy vs Palliative treatment/support (*p* = 0.614 and *p* = 0.447). Significant *p*-values are indicated as bold numbers.

**Table 4 biomolecules-12-00935-t004:** Univariate analysis of cancer-specific survival (CSS) according to clinical and histopathological variables.

Factors	N	Dead	CSS 1-Year (%)	CSS 2-Years (%)	CSS Mean ± S.D. (CI 95%)	*p*-Value
**Gender**						0.202
Female	22	12	43.4	32.5	22.60 ± 6.49 (9.88–35.33)	
Male	28	21	21.1	15.8	6.59 ± 1.73 (3.19–9.98)	
**Age (years)**						0.498
<7 year-old	5	2	60	60	20.49 ± 7.4 (5.99–34.99)	
≥7 year-old	45	31	28	18.4	14.31 ± 3.86 (6.74–21.88)	
**Breed**						0.247
Small	8	4	34.3	34.3	9.72 ± 3.78 (2.30–17.14)	
Medium	3	1	66.7	66.7	11.32 ± 3.81 (3.83–18.81)	
Large	10	8	20	20	7.70 ± 4.16 (0–15.86)	
UB	22	14	28.7	28.7	14.73 ± 5.73 (3.50–25.95)	
**Tumour Location**						0.643
Mouth (NOS)	7	4	33.3	33.3	9.01 ± 3.17 (2.79–15.23)	
Gingiva	18	9	40	40	24.98 ± 7.34 (10.58–39.38)	
Tongue	10	8	25	0	6.26 ± 2.69 (0.98–11.52)	
Oropharynx (including Tonsils)	8	7	0	0	4.02 ± 1.09 (1.89–6.16)	
Palate	7	5	28.6	28.6	8.39 ± 3.98 (0.60–16.18)	
**Histological type**						**0.013**
Papillary SCC	6	1	80	80	48.3 ± 9.3 (30.07–66.53)	
Conventional SCC	44	32	24.1	13.4	6.94 ± 1.43 (4.13–9.75)	
**Bone Invasion**						0.856
Yes	8	4	35	35	21.35 ± 11.33 (0–43.56)	
No	42	29	30.4	20.2	10.45 ± 2.20 (6.14–14.77)	
**Vascular Invasion**						0.689
Yes	4	2	50	50	11.71 ± 5.29 (1.34–22.09)	
No	46	31	29.6	19.8	15.08 ± 3.97 (7.30–22.87)	
**Anneroth´s histological grade**						0.543
Well differentiated	16	10	33.9	33.9	21.55 ± 6.89 (8.04–35.06)	
Moderate differentiated	34	23	28.7	16.4	7.97 ± 1.75 (4.55–11.40)	
Poor differentiated	0	-	-	-	-	
**Bryne’s histological grade**						0.112
Well differentiated	22	16	55.8	16.7	12.20 ± 4.88 (2.63–21.77)	
Moderate differentiated	26	15	44.7	26.1	10.95 ± 2.21 (6.62–15.28)	
Poor differentiated	2	2	0	0	1.35 ± 0.15 (1.06–1.64)	
**Mitosis number**						0.934
0–1/hpf	16	11	31.3	31.3	20.19 ± 6.51 (7.43–32.94)	
2–3/hpf	15	10	23.5	0	6.58 ± 2.28 (2.10–11.05)	
4–5/hpf	13	7	46	34.5	13.85 ± 4.75 (4.54–23.16)	
>5/hpf	6	5	16.7	16.7	6.88 ± 3.41 (8.53–23.65)	
**Nuclear Pleomorphism**						0.069
Few	8	3	57.1	57.1	21.54 ± 5.47 (10.82–32.25)	
Moderately	22	14	32.3	24.2	16.82 ± 5.88 (5.28–28.36)	
Abundant	20	16	17.7	8.8	5.43 ± 1.75 (2–8.87)	
Lymphocytic infiltration						0.448
Weak	14	9	20	10	6.18 ± 1.87 (2.51–9.85)	
Moderate	21	11	40.3	32.3	21.56 ± 6.66 (8.51–34.61)	
Strong	15	13	31	31	6.33 ± 1.95 (2.51–10.15)	
**Pattern of invasion**						**0.011**
I	15	5	61.9	61.9	37.74 ± 7.40 (23.24–52.25)	
II	19	16	13	0	4.79 ± 1.50 (1.84–7.74)	
III	13	10	20	0	4.15 ± 1.44 (1.33–6.97)	
IV	3	2	33.3	33.3	7.84 ± 5.90 (0–19.41)	
Stage of invasion						**0.009**
I	0	-	-	-	-	
II	31	17	42.9	25.0	19.58 ± 5.77 (8.27–30.88)	
III	18	15	12.3	12.3	4.75 ± 1.90 (1.01–8.48)	
IV	1	1	0	0.00	76 ± 0 (76–76)	
**Treatment**						**0.048**
Surgery	11	4	55.6	55.6	20.74 ± 4.95 (11.03–30.44)	
Chemotherapy	4	14	0	0	3.46 ± 1.60 (0.33–6.58)	
Palliative treatment/support	35	25	26.9	14.4	11.97 ± 4.12 (3.89–20.05)	
**Tumour stage**						**0.001**
I + II	14	5	64.6	48.5	32.53 ± 8.61 (15.66–49.40)	
III + IV	30	24	9.9	9.9	3.45 ± 0.74 (2.00–4.89)	

Legend: UB, undetermined breed; NOS, not otherwise specified; SSC, squamous cell carcinoma; hpf, high power field; significant *p*-values are indicated as bold numbers.

**Table 5 biomolecules-12-00935-t005:** Univariate analysis of cancer-specific survival (CSS) according to biomarkers expression.

Factors	N	Dead	CSS 1-Year (%)	CSS 2-Years (%)	CSS Mean ± S.D. (CI 95%)	*p*-Value
**p-mTOR extent (% of tumour cells)**						0.143
0–49%	13	6	43.9	43.9	27.13 ± 8.84 (9.81–44.45)	
50–100%	37	27	26	10.8	8.14 ± 2.12 (3.99–12.29)	
**p-mTOR intensity**						0.290
Negative/weak	6	3	41.7	41.7	15.54 ± 7 (1.82–29.27)	
Moderate/strong	44	30	28.6	28.6	13.65 ± 4.01 (5.62–21.68)	
**p-mTOR location**						0.699
Cytoplasm	4	3	25	25	7.07 ± 5.18 (0–17.22)	
Membrane + cytoplasm	43	27	30.7	23	16.87 ± 4.43 (8.19–25.56)	
Cytoplasm + nucleus	3	3	33.3	0	5.353 ± 3.66 (0–12.52)	
**p-mTOR distribution**						0.255
Central	7	2	64.3	64.3	38.89 ± 11.59 (16.18–61.6)	
Homogeneous	40	29	76.6	52	9.27 ± 2.15 (5.06–13.47)	
Periphery	3	2	0	0	5.02 ± 3.19 (0–11.26)	
**p-S6 extent (% of tumour cells)**						**0.023**
0–24%	8	2	62.5	62.5	38.42 ± 11.50 (15.88–60.96)	
25–100%	42	31	24.7	14.1	8.32 ± 2.01 (4.38–12.26)	
**p-S6 intensity**						0.325
Negative/weak	6	3	33.3	33.3	22.41 ± 13.645 (02.95–49.16)	
Moderate/Strong	44	30	29.3	19.6	9.96 ± 2.17 (5.71–14.21)	
**p-S6 location ***						0.565
Cytoplasm	24	17	33.3	18.5	10.6 ± 2.77 (5.16–16.03)	
Membrane + cytoplasm	25	16	25.2	25.2	16.43 ± 5.43 (5.79–27.07)	
**p-S6 distribution ***						0.157
Central	5	2	50	50	11.13 ± 3.67 (3.93–18.34)	
Homogeneous	37	27	23	12.8	6.68 ± 1.54 (3.66–9.69)	
Periphery	7	4	42.9	42.9	26.72 ± 10.49 (6.17–47.28)	

Significant *p*-values are indicated as bold numbers. * includes a negative case without any expression on tumour cells and because of that excluded in this analysis.

**Table 6 biomolecules-12-00935-t006:** Multivariate analysis of the cancer-specific survival.

	Cancer-Specific Survival	
Variables	*p*-Value	HR (95% CI)
**Histological type**		
Papillary	**0.938**	1 (reference category)
Conventional		1.1 (0.11–10.86)
**Treatment**		
Surgery or chemotherapy	**0.009**	1 (reference category)
Palliative or support		1.95 (1.18–3.22)
**p-S6 extent**		
0–24%	**0.027**	1 (reference category)
25–100%		11.58 (1.316–101.83)
**Tumour stage**		
I + II	**0.013**	1 (reference category)
III + IV		5.01 (1.41–17.83)
**Pattern of invasion**		
I	0.059	1 (reference category)
II	**0.016**	6.72 (1.43–31.59)
III	**0.007**	10.53 (1.92–57.87)
IV	0.05	10.52 (1–111.24)
**Stage of invasion**		
I	-	-
II	0.661	1 (reference category)
III	0.421	1.47 (0.58–3.73)
IV	0.522	2.1 (0.22–20.29)

HR, hazard ratio; CI, confidence interval for HR. Variables included in multivariable Cox regression analysis using enter method. Significant *p*-values are indicated as bold numbers.

## Data Availability

The data information could be asked to the correspondent author.

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
