# Peer review of "p-S6 as a Prognostic Biomarker in Canine Oral Squamous Cell Carcinoma"

_biomolecules, 2022, doi:10.3390/biom12070935_

Round 1

Reviewer 1 Report

The authors studied the protein expression of phospho-mTOR (p-mTOR) and one of its downstream targets phospho-S6 (p-S6) in dogs with oral squamous cell carcinoma and explored the association of their expression with several histopathological and clinical parameters. They found an association between p-mTOR expression and tumor invasion and number of mitoses as well as between p-S6 expression and necrosis. Also, tumor stage, invasion, histological subtype, treatment, and p-S6 expression were significantly associated with an increased overall survival.

The study is novel and interesting for the field. The authors put a lot of efforts in analyzing all these data. However, there are a few major critical points that require authors’ attention and need to be addressed before the manuscript can be considered suitable for publication.

MAJOR CONCERNS

1.     Antibody specificity needs to be evaluated when working with antibodies that have not been tested on the species of interest. None of the antibodies is claimed to work in dogs by the company. Please provide a western blot showing antibody specificity on canine samples with appropriate controls. Alternatively, if the same antibodies were used in other studies, please include this information in the text with the appropriate citation.

2.     Cut-off values used when evaluating p-mTOR and p-S6 expression are different (lines 165-166). Can the authors explain why they did not use the same cut-off values, particularly considering that they studied the correlation between the two? Could this be the reason why there was no correlation between intensity of their expression, as stated?

3.     Some of the associations found between protein expression and histopathological parameters (tumor invasion, mitosis, and necrosis) are biologically contradictory. P-mTOR and p-S6 are considered markers of aggressiveness. One would expect their expression should positively correlates with aggressiveness. Here, the authors found that the likelihood of having a high p-mTOR expression (extent) decreases when the pattern of invasion increases from I (2low vs 18high) to II, III, IV (15low vs 26high). Similarly, the likelihood of having a high p-mTOR expression (intensity) decreases when the number of mitoses increases. Yet, the likelihood of having a high p-S6 expression (intensity) increases when there is NO necrosis, which is often an indication of a more benign tumor (necrosis is indeed often associated with high aggressiveness).

Aren’t these results in contrast with the hypothesis that p-mTOR and p-S6 are markers of an activated mTOR signaling, and therefore of aggressiveness?

MINOR POINTS

1.     Throughout the manuscript, the protein S6 is also called S6K1. Please always use the same name when referring to the same protein.

2.     Line 24-25: “…were present in almost cases.” . I believe the authors meant “…were present in almost all cases”. I noticed this mistake is repeated multiple times. Please modify.  

3.     Immunoreactivity was evaluated based on the extent and the intensity of staining. Since extent referes to percentage of positive cells, please change the term “extent” with “percentage” throughout the whole manuscript. This would clarify how the evaluation was performed as I believe “extent” of staining is not commonly used.

4.     Figures 1 and 2 are not cited in the text. Please cite them. Also, figures 1 and 2 are blurry and could not be evaluated. Would it be possible to upload a high quality version? Additionally, please add a scale bar in figures 1 and 2.

5.     In figure 1 legend, the authors stated that figure A3 and D3 show a high expression of p-mTOR. To me, the intensity in figure A3 looks more similar to the intensity in figure C3, which is described as low intensity. Also, would it be possible to order the single cases of figures 1 and 2 from low-to-high intensity? For example, in figure 2, I would put B before A, C, and D.

6.     The authors stated some of the samples showed a membrane expression of these proteins. Is this expected? Aren’t these proteins expressed only in the cytoplasm? Please clarify.

7.     Table S1 is cited but I believe was not uploaded.

8.     Tables 4 and 5. Please add asterisks to all significant p-values (for example, 0.009 is significant but there is no asterisk).

9.     Table 5. Are values referring to CSS 1-year and CSS 2-year percentages? Please specify.

10.  Please add p-values to figures 3 and 4 (Kaplan-Meier curves).

11.  Use of English language is sometimes not appropriate. Some of the sentences, especially in the discussion section, are hard to follow and therefore need to be rephrased.

Reviewer 2 Report

In the manuscript by Delgado et al the authors describe an immunohistochemical study of p-S6 in canine oral cancer (n=61). The study appears robust and well executed and the methods appropriate. Some of the aspects should be explained or further elaborated.

Analysis and data

P7 L223 why was pS6 expression binarized as 0-24% and 25-100% while mTOR was binarized as 0-49 and 50-100%?  Binarizing pS6 in groups of 8 and 53 cases makes the groups very uneven and statistics difficult. Would the results change if both proteins were analyzed in the same way?

Considering that the authors performed a large number of independent statistical tests (>70 chi square tests), was any thought given to multiple testing corrections to alleviate false positive results?

P10 Table 4 the data for males and females is identical for the CSS mean which is likely a data entry problem. Age variable also has data issues in the same column

P11 Table 5 p-S6 location and pS6 distribution rows have only 49 cases instead of the expected 50 cases with survival information? Please doublecheck the information provided/analyzed.

Is it possible to combine the extend and intensity scores to make a combined classifier?

Clarity

P2L84 INNO is likely an acronym and should be spelled out at least at first mention.

P5 Table 1. Breed is listed as having only 53 cases (of 61) with 26 listed as UB (likely standing for unknown breed). However, the table should have 61-53 = 8 additional cases of unknown breed as well. Or the star”*” should further explain the variable in the table footer. Possibly the unknown (UB) should stand for mixed breed, while unknown should be reserved for completely missing data.

P10/11 Tables 4 and 5. P-value has a cross indicator suggesting further information in the table footer, However, the footer doesn’t contain any explanations for this sign.

P10 Table 4 “4-5/hpf “ mitosis row has missing decimal number in CSS 1-year column “46.”

P10/11 Tables 4 and 5 Consider rounding all CSS values to 1 decimal point.

P12 Table 6 consider rounding all HR values to 1 or 2 decimal points (in the text as well). 3 is excessive except for p values. Double check rounding since row IV (pattern of invasion) is listed as having p=0.05 but the confidence interval crosses 1 which usually means the difference is not significant possibly p is 0.051 which is lost if only this value is rounded to 2 decimal points.

Since Tables and Figures depicting survival analysis are often not on the same page, consider including the relevant P values in Kaplan Meier graphs to facilitate reading

P13 L 302 The sentence should be clarified and or rewritten “To our knowledge on association of p-mTOR expression and clinical and pathologic variables in canine oral cancer exist for far to our knowledge, which difficult comparisons of the present results.”

Language and typos

P1 L25 „in almost cases.“

P2 L75 no space „human cancers[24, 29-31].”

P2 L94 no comma ”histopathological grade presence”

Paragraph at P7 L200-207 has several spacing typos

P7 L221 “almost of the cases (60; 98.4%)”

Paragraph at P9 L233-239 has several spacing typos

P13 L 285 typo “of these marker”

P13 L 292 typo “is being direct to mTOR”

Round 2

Reviewer 1 Report

Dear authors,

Thank you for providing your responses to my comments. I still have a few suggestions that, in my opinion, would improve this manuscript.

In the abstract, please modify the term “management” with “treatment” (line 28).

In the methods, please indicate how many cells (total and per field) were counted to calculate the percentage/extent of labeling.

Chi-square test results of p-mTOR/p-S6 expression and clinico-pathological variables should be described differently, as they are now misleading. As previously mentioned, the significant associations between p-mTOR/p-S6 expression and tumor invasion/mitosis/necrosis are somehow biologically contradictory. Please describe/interpret these results more clearly in the result section. Right now, lines 213-216 read “(…) a significant positive association was observed between p-mTOR extent score with pattern of tumour invasion when categorized by I vs II + III + IV grades and between p-mTOR intensity score expression and number of mitosis.” I am not sure the term “positive” association can be used for this type of analysis, as it is meaningless (this is not a correlation!). So, please remove the term “positive”. Also, please add a sentence that gives a clearer interpretation of these results. If I understand it correctly, the authors could also write something like “Tumors with a high pattern of invasion were less likely to have a high p-mTOR extent score. Also, tumors with a high number of mitoses were less likely to have a high p-mTOR intensity score.”

Similarly, lines 244-245 read “A significant positive association was observed between p-S6 intensity expression and presence of necrosis.” Please remove “positive” and add something like “Tumors with necrosis were less likely to have a high p-S6 intensity score.”, to give an interpretation of these results.

Lines 266 and 378: please modify “correlated” with “associated”, as I believe no correlation analysis was done in this study.

Table 5. Please modify the column titles “CSS 1-year” and “CSS 2-year” with “CSS 1-year (%)” and “CSS 2-year (%)”, respectively (like in table 4).

I cannot see the original figure files. I only see those embedded in the text and they are very blurry. Please make sure the final figures are not blurry.

One of the conclusions of this study is that p-mTOR is deregulated in OSCC. I wonder which data support this conclusion, considering that essentially all the analysis on p-mTOR expression were not significant.

Additionally, this manuscript requires an extensive editing of English language, which is sometimes hard to follow. 

Reviewer 2 Report

The authors have addressed all comments sufficiently.

Regarding point 1, i understand the reasoning behind the using ROC curves to establish cutoffs. My original comment was not focusing on the cutoff itself but on the very uneven groups that are established with this cutoff. Comparing 8 to 53 cases is often problematic as random chance plays a major role in the 8 cases group. 

However, if the authors already did the analysis on different cutoffs and the results are the same then my concern is alleviated.
